# Arginine Derivatives in Cerebrovascular Diseases: Mechanisms and Clinical Implications

**DOI:** 10.3390/ijms21051798

**Published:** 2020-03-05

**Authors:** Gerrit M. Grosse, Edzard Schwedhelm, Hans Worthmann, Chi-un Choe

**Affiliations:** 1Department of Neurology, Hannover Medical School, 30625 Hannover, Germany; worthmann.hans@mh-hannover.de; 2Institute of Clinical Pharmacology and Toxicology, University Medical Center Hamburg-Eppendorf, 20249 Hamburg, Germany; schwedhelm@uke.de; 3DZHK (Deutsches Zentrum für Herz-Kreislauf-Forschung e.V.), partner site Hamburg/Kiel/Lübeck, 20249 Hamburg, Germany; 4Department of Neurology, University Medical Center Hamburg-Eppendorf, 20249 Hamburg, Germany; cchoe@uke.de

**Keywords:** ADMA, atherosclerosis, arginine, atrial fibrillation, biomarker, endothelial dysfunction, ESUS, homoarginine, SDMA, stroke

## Abstract

The amino acid L-arginine serves as substrate for the nitric oxide synthase which is crucial in vascular function and disease. Derivatives of arginine, such as asymmetric (ADMA) and symmetric dimethylarginine (SDMA), are regarded as markers of endothelial dysfunction and have been implicated in vascular disorders. While there is a variety of studies consolidating ADMA as biomarker of cerebrovascular risk, morbidity and mortality, SDMA is currently emerging as an interesting metabolite with distinct characteristics in ischemic stroke. In contrast to dimethylarginines, homoarginine is inversely associated with adverse events and mortality in cerebrovascular diseases and might constitute a modifiable protective risk factor. This review aims to provide an overview of the current evidence for the pathophysiological role of arginine derivatives in cerebrovascular ischemic diseases. We discuss the complex mechanisms of arginine metabolism in health and disease and its potential clinical implications in diverse aspects of ischemic stroke.

## 1. Precision Stroke Medicine: on Search for Novel Biomarkers

Stroke is globally the second leading cause of death and morbidity [1]. While stroke-associated mortality decreased between 1990 and 2010, stroke prevalence, incidence as well as mortality rates again raised between 2010 and 2017 [2], despite optimized treatment options and intervention programs. Moreover, stroke burden is also increasing in young adults [3]. According to recent findings from the Global Burden of Disease study, the life-time risk to suffer stroke is about 25% starting at the age of 25 years [3]. Facing the challenges of this global cerebrovascular disease epidemic the need of biomarkers supporting individual stroke patient care in terms of precision medicine is becoming increasingly evident [4,5]. This holds true for estimating the individual risk for cerebrovascular diseases for primary preventive strategies but also for secondary prevention after the event. Stroke is a complex disease of diverse underlying risk factors and etiologies and current evidence underscores that a thorough individualized investigation of these conditions is needed for the purpose of an optimal treatment [4]. Thus, there are intense efforts in identifying appropriate imaging, genetic or blood biomarkers that are able to reflect the underlying pathophysiology and are useful for clinical decision making. L-arginine (Arg) derivatives may meet the conditions of such clinically interesting targets in cerebrovascular diseases. For this narrative review article, we conducted a comprehensive literature search through PubMed and selected original articles, reviews and meta-analyses on Arg and its derivatives in ischemic stroke, underlying risk factors and etiological diseases. We aim to provide an overview of the current evidence on how the Arg metabolism is involved in cerebrovascular pathophysiology and how Arg derivatives may constitute valuable biomarkers of risk, morbidity and mortality as well as etiology of ischemic stroke.

## 2. Metabolism of Arginine and its Derivatives 

The amino acid Arg is synthesized in the kidney via the urea cycle. Cells which are not synthesizing Arg alone are able to take up Arg via cationic amino acid transporters (CAT). Arginine serves as substrate of the nitric oxide (NO) synthase which exists in three isoforms: the inducible NOS (iNOS), the neuronal NOS (nNOS) and the endothelial NOS (eNOS) [6]. Homoarginine (hArg), which is structurally similar to Arg, primarily originates from the catalytic activity of arginine:glycine amidinotransferase (AGAT) but probably urea cycle enzymes are also involved [6]. Homoarginine serves—although with low affinity—as a substrate of NOS and, moreover, may lead to an increased bioavailability of Arg by inhibiting the enzyme arginase. Further endogenous derivatives of Arg are symmetric dimethylarginine (SDMA), asymmetric dimethylarginine (ADMA) and monomethylarginine (NMMA). At first, proteins are methylated by protein arginine methyltransferases (PRMT) I or II. While PRMT I catalyzes methyl groups asymmetrically, PRMT II leads to a symmetric arrangement [7]. Subsequently, methylarginine residues are released during proteolysis of proteins with methylated arginine residues. ADMA and NMMA are endogenous NOS inhibitors competing with Arg and hArg. In contrast, SDMA was identified as inhibitor of cellular Arg uptake via CAT. As a result, methylarginines lead to a deprivation of the bioavailability of NO which is a key endogenous regulator of vascular tone, angiogenesis, inhibition of platelet activation as well as leukocyte adhesion [8,9]. Nitric oxide moreover leads to a decreased endothelial expression of monocyte chemoattractant protein 1 (MCP-1) [10] and further leukocyte adhesion molecules [11]—a mechanism which is therefore regarded as protective in early stages of atherosclerosis [8]. Proliferation of vascular smooth muscle cells and thus formation of fibrous plaque which is a hallmark of further established atherosclerotic lesions is also inhibited by NO [8,12]. Abnormal neuronal NO signalling has been implicated in neurodegenerative disorders like Alzheimer’s and Parkinson’s disease and also in neurodegeneration following stroke [13].

Furthermore, when the bioavailability of Arg is reduced, a shift of the enzymatic activity of NOS may occur, resulting in the production of superoxide anions (radical oxygen species, ROS)—often referred as “NOS uncoupling” [14]. Further NO inactivation may be caused by its reaction with the superoxide anion resulting in formation of peroxynitrite which is a potent oxidant causing damage of proteins, lipids and DNA [8]. Generation of vascular superoxide was correlated with ADMA in patients with coronary heart disease [15]. Feliers et al. [16] reported that also SDMA may cause eNOS uncoupling in glomerular endothelial cells. Thus, both dimethylarginines are not only associated with decreased bioavailability of NO but may also contribute to increased ROS-production [15,16]. Reactive oxygen species are crucial in the progression of atherosclerosis [17]. Vice versa, states of oxidative stress like inflammation or cellular damage, as for example induced by stroke, themselves lead to augmentation of dimethylarginine production via supporting PRMT activity and inhibiting the dimethylarginine dimethylaminohydrolase (DDAH) [18] (see paragraph 4). DDAH is the enzyme degrading ADMA and NMMA, existing in two isoforms, being expressed constitutively [19]. An alternative minor elimination route is the mitochondrial alanine-glyoxylate aminotransferase 2 (AGXT2) which is primarily located in the kidney and metabolizes not only ADMA but also to some extend SDMA [20]. Both, ADMA and in particular SDMA, are excreted via the kidney unmetabolized, explaining high correlations with renal function [21]. However, the pathological relevance of free ADMA and SDMA remains controversial given the fact that the free forms of dimethylarginines comprise only weak inhibitory potency towards NOS [22]; nevertheless, there is mounting evidence of Arg derivatives as promising targets in cerebrovascular diseases which will be discussed below (Figure 1).

## 3. Arginine Derivatives as Markers of Cerebrovascular Risk and Mortality

### 3.1. The Relation of ADMA and SDMA to Atherosclerotic Disease

As discussed above, dimethylarginines are associated with reduced bioavailability of NO, either via competing with Arg at the catalytic active side of NOS (ADMA) or via inhibition of the cellular Arg uptake by SDMA. Dimethylarginines are therefore regarded as markers of endothelial dysfunction which is an early step in the initiation of atherosclerosis [23]. The first evidence for ADMA as a risk marker in stroke patients came from Korean researchers investigating plasma ADMA concentrations in elderly patients [24]. Plasma ADMA was determined with high-performance liquid chromatography and fluorescence detection (HPLC-FL) and found to be doubled in 52 stroke patients as compared with 35 age and sex-matched controls. More importantly, the odds ratio (OR) for recurrent stroke for the upper median ADMA level of total subjects was even seven-fold increased. A few years later, increased concentrations of ADMA were reported for hemorrhagic and cardio-embolic stroke as well as transient ischemic attack (TIA) patients in the Swedish population of Kalmar [25].

The functional relevance of ADMA has been proven by Kielstein et al. showing that infusion of ADMA in healthy adults does not only lead to higher arterial resistance and blood pressure [26] but also to decreased cerebral perfusion [27] which underscores the role of ADMA as active mediator rather than a mere biomarker of endothelial dysfunction. In accordance, Baum et al. [28] reported significant associations between Arg derivatives with measure of vascular function in 5.000 individuals. The pathophysiological impact of Arg derivatives in atherosclerosis is furthermore supported by experimental work showing that ADMA is associated with foam cell formation [29] as well as with migration [30] and apoptosis [31] of vascular smooth muscle cells [30]. Interestingly, ADMA suppresses endothelial progenitor cells (EPC) in patients with coronary heart disease [32]. EPC play an important role in the regeneration of injured endothelium [32]. 

Progression and vulnerability of atherosclerotic lesions is driven by inflammatory mechanisms [33]. ADMA may be increased by systemic inflammation and subsequently lead to endothelial dysfunction in patients with coronary heart disease or rheumatoid arthritis (RA) [34]. The link of ADMA and inflammation in RA patients is meanwhile confirmed by different studies and meta-analyses [35,36,37] suggesting that ADMA may constitute as vascular risk marker in this vulnerable patient collective. This is further supported by a study in RA patients demonstrating an association of ADMA with homocysteine and with a methylenetetrahydrofolate reductase (MTHFR) polymorphism which are implicated in atherosclerosis [38].

Carotid intima media thickness (CIMT) is a marker of subclinical atherosclerosis and regarded as predictor of vascular diseases [39]. While one study described an inverse relation of ADMA with CIMT [40], there are meanwhile numerous studies proving a positive correlation between circulating ADMA and/or SDMA concentrations with CIMT in patients with diverse demographics and different underlying diseases [41,42,43,44,45,46,47,48,49,50,51,52,53,54,55,56], however, with ethnical differences [57]. In a long-term follow-up study over six years, Furuki et al. [47] showed that ADMA was an independent predictor of CIMT progression. Interestingly, in an analysis of the Framingham offspring cohort, an independent association of ADMA with CIMT in the internal carotid artery (ICA) was confirmed but not with that in the common carotid artery (CCA), implying a site-specific role of ADMA in atherosclerotic disease [52]. Of note, CIMT at the ICA might be the superior measure of cardiovascular risk compared with CIMT at the CCA [39]. In a study including patients with recent ischemic stroke or TIA an association between hArg/ADMA ratio and the aortic intima media thickness has been reported [58].

The relevance of dimethylarginines in stroke due to large artery atherosclerosis will be discussed in paragraph 5.1.

### 3.2. ADMA and SDMA in Relation to Vascular Risk Factors

Hypertension is the most important stroke risk factor [59]. Nitric oxide deficiency as well as oxidative stress play an important role in arterial hypertension. Likewise, numerous investigations have shown a link between ADMA levels and the occurrence and development of hypertension [60,61,62]. Controversially, there are also studies in the general population which could not prove this association [63]. Of note, chronic SDMA infusion in otherwise healthy mice did not lead to an alteration of blood pressure, concluding that SDMA is unlikely to be a causal pathophysiological factor [64]. 

ADMA was found to be highly elevated in young patients with hypercholesterolaemia [65]. A potential mechanistic link might be based on the upregulation of ADMA synthesis through LDL cholesterol [66]. Vice versa, in a preclinical hypercholesterolemia model the reduction of ADMA levels was related to less atherosclerotic lesions [67]. Interestingly, in a study including 3.310 patients undergoing coronary angiography, SDMA was identified as marker of high density lipoprotein (HDL) dysfunction, markedly in patients with renal insufficiency which may implicate another pathophysiologic link between renal disease, SDMA and premature vascular disorders [68]. In the original study by Yoo et al. [24] ADMA was moreover found to be positively correlated with homocyst(e)ine.

Importantly, in patients with embolic stroke of undetermined source (ESUS) who often reveal only few vascular risk factors [69] a strong correlation between the CHA_2_DS_2_VASC and both dimethylarginines, especially with SDMA, was found [46]. CHA_2_DS_2_VASC has been previously shown to be a reliable instrument predicting recurrent cerebral ischemia and death in ESUS patients [70]. An association between CHA_2_DS_2_VASC and both dimethylarginines as well as hArg/dimethylarginine ratios has been recently proven in three independent cohorts of patients with cerebrovascular diseases [71]. 

Nitric oxide released from endothelial cells strongly inhibits platelet activation and vascular adhesion [72,73]. In patients undergoing percutaneous angioplasty ADMA was shown to be associated with platelet activation. In accordance, in hypertensive patients higher ADMA levels may lead to increased platelet aggregation via cGMP signalling [74,75]. Thus, Arg derivatives may potentially play a role also in platelet activation during stroke.

### 3.3. Dimethylarginines Predict Morbidity and Mortality of Cerebrovascular Diseases

In a meta-analysis by Willeit et al. [76] including 19,842 individuals, those with ADMA levels in the highest tertile compared with those in the lowest tertile showed significantly higher risk for cardiovascular diseases (hazard ratio, HR 1.42 (1.29–1.56)), coronary heart disease (HR 1.39 (1.19–1.62)) and especially for stroke (HR 1.60 (1.33–1.91)). However, in the same data set, SDMA levels did not show a significant association with these vascular outcome measures (HR for stroke: 1.31 (0.83–2.07)) [76]. As described above, the relevance of SDMA as risk factor for vascular endpoints might be higher in patients who suffer from renal insufficiency [77,78]. In a sub-study of the ARISTOTLE trial, Horowitz et al. [79] investigated ADMA and SDMA in anticoagulated patients with atrial fibrillation (AF). While ADMA concentrations were weakly related to thromboembolic events, the authors found an association of SDMA with bleeding events and of both dimethylarginines with increased mortality [79]. 

In total, there are far less studies focussing on SDMA compared with ADMA [18,80]. As described above, SDMA is an excellent marker of renal function which itself is related with vascular diseases [81]. Though, there is evidence that SDMA reflects vascular risk and disease independent from the extent of renal insufficiency [82]. Interestingly, Emrich et al. [77] demonstrated in a study of 528 patients with chronic kidney disease (CKD) that SDMA was superior to other Arg derivatives in predicting CKD progression as well as vascular events including stroke. Two independent prospective studies showed that SDMA predicts short- and long-term outcome following ischemic stroke [83,84]: Lüneburg and colleagues showed in 137 acute ischemic stroke patients who were admitted to the emergency unit that plasma SDMA was associated with composite detrimental outcome as comprised as death, recurrent stroke, myocardial infarction and re-hospitalization in the first 30 days after the event—an association which was mediated by the link of SDMA and renal function [84]. In the other study, survival was independently related to lower SDMA during a median time of follow-up of 7.4 years in 394 patients with ischemic stroke while ADMA barely missed significance [83]. Thus, SDMA is currently emerging as an interesting target in cerebrovascular diseases. Table 1 and Table 2 provide an overview on the discussed clinical studies investigating ADMA and SDMA in cerebrovascular diseases.

### 3.4. Homoarginine as Marker and Target in Cerebrovascular Diseases

A decade ago, hArg was studied in regard to cardiovascular and all-cause mortality [89]. In contrast to dimethylarginine derivatives, hArg levels were inversely associated with adverse events and mortality [90]. Most consistently, low hArg levels are associated with all-cause and cardiovascular mortality, which was shown in subjects referred for coronary angiography, in hemodialysis patients with diabetes mellitus, in stroke patients, but also population-based cohorts [89,91,92,93,94]. A recent meta-analysis confirmed the inverse association of hArg with all-cause mortality (HR 0.64 [0.57–0.73]) [95]. More specifically, low hArg levels were strongly associated with fatal strokes and revealed a trend to increase stroke risk [96,97]. In prospective studies of stroke patients, low hArg levels were independently associated with increased long-term all-cause mortality and short-term adverse events, respectively (Table 3) [91]. In addition to cerebrovascular patients, increased hArg levels (i.e., 1-SD log plasma hArg) were also associated with a risk reduction for major adverse cardiovascular events including stroke [93]. See Table 3 for an overview on clinical studies evaluating hArg as biomarker in cerebrovascular diseases.

Homoarginine has been implicated to play a role in vascular function [90,99,100,101]. Correspondingly, epidemiological studies revealed an inverse association of hArg with aortic wall thickness and aortic plaque burden [93], an inverse correlation of hArg/ADMA ratio with aortic intima-media thickness [58] and a link between hArg/SDMA ratio with internal carotid artery stenosis and unfavorable outcome after stroke [71,98].

Most importantly, hArg supplementation confers indeed beneficial effects in vascular disease models. First, AGAT-deficient mice are devoid of hArg and revealed increased infarct sizes and an impaired cardiac contractibility, which were normalized upon hArg supplementation [91,102]. Furthermore, hArg supplementation attenuated detrimental effects of diabetic kidney damage, preserved systolic function in a model of coronary artery disease, reduced neointimal hyperplasia in balloon-injured rat carotids and attenuated post-myocardial infarction heart failure [91,103,104,105,106]. 

Although mouse studies revealed a causal link between hArg and vascular disease, the direct protective effect in humans remains to be established [107]. In a recent clinical trial, pharmacokinetic and -dynamic parameters were studied in healthy volunteers orally supplemented with 125mg hArg or placebo daily for 4 weeks using a cross-over design [108]. Supplementation was well tolerated and increased hArg levels by 7 fold without any alteration of vascular or neurological parameters [108,109]. Currently, a randomized placebo-controlled trial studies the administration of oral hArg in acute stroke patients with low hArg levels (https://www.clinicaltrials.gov; unique identifier: NCT03692234). Future studies will answer the question, if hArg deficiency is a modifiable protective cerebrovascular risk factor.

## 4. Acute Response of Arginine Derivatives after Ischemic Stroke

After the acute onset of stroke, the local reaction to the ischemic tissue based on cellular damage, proteolysis and oxidative stress consecutively induces a secondary systemic reaction. In this setting, an increase in expression of PRMTs and for ADMA a decreased activity of DDAH elevates systemic levels of dimethylarginines. More extended acute brain damage in larger cerebral infarction accelerates cellular damage and proteolysis resulting in further increase of dimethylarginines. In patients with acute ischemic stroke several studies demonstrated the increase of dimethylarginines; while studies including ADMA are numerous, those including SDMA are sparse.

Brouns and colleagues [85] reported dimethylarginine concentrations in 88 acute ischemic stroke and TIA patients within 24 hours of onset. This study is remarkable in several aspects. Firstly, ADMA as well as SDMA levels increased with increased severity as assessed by the National Institutes of Health stroke scale (NIHSS). Secondly, significant correlations between outcome, evaluated by the modified Rankin scale (mRS) at three months after stroke, and both dimethylarginine derivatives existed and third, ADMA and SDMA were measured in cerebrospinal fluid (CSF) within 24 hours of stroke onset. This indicates that elevated ADMA and SDMA in the hyperacute phase of stroke originate from cerebral cellular damage and proteolysis. 

The kinetics during the acute phase of stroke seem to be different between Arg, SDMA and ADMA. ADMA seems to peak rather early within hours after admission to the hospital while SDMA increases and L-arginine decreases during the first days after admission [86,88,110]. Both, the time courses of ADMA and SDMA relate to the clinical outcome assessed by mRS, while SDMA and in particular Arg, moreover, independently predict post-stroke infections [86,110]. The kinetics observed in stroke patients are similar to those seen in septic patients, and also these Arg derivatives may help distinguish between patients with favorable from unfavorable outcome [111,112].

Higher ADMA levels were also detected in 40 Chinese TIA patients compared to controls [113]. In contrast to this, in a Spanish study there was no difference in ADMA levels in 238 ischemic stroke patients and controls [87]. However, stroke etiology or time of blood withdrawal was not indicated in this study. The authors suggested genetic, socio-economic or nutritional factors explaining differences to other stroke patient cohorts. In 52 Turkish patients with acute ischemic stroke ADMA levels were higher than in controls, while NO levels measured in stroke patients were lower than in controls [114]. One might wonder if ADMA increase is directly linked to the extent of acute brain injury. This remains unclear, since no correlation of ADMA and S100B concentrations has been found in 58 ischemic stroke patients [115] but a weak correlation of ADMA, SDMA and S100B has been shown in another study in 55 ischemic stroke patients [88].

There is less data about SDMA after ischemic stroke, since initially greater importance for stroke pathophysiology was attributed to ADMA due to its role as endogenous NOS inhibitor. Meanwhile also SDMA levels have been repeatedly measured, showing that these are also elevated after the acute event of stroke. In 55 acute ischemic stroke patients, SDMA levels at 6 hours were associated with neurological worsening [110]. In another study, 67 acute ischemic stroke patients with unfavorable outcome showed an elevation as early as 6 hours until 3 days after the event of stroke [86]. Brouns et al. [85] demonstrated increased SDMA levels in CSF, while this increase was pronounced in more severe stroke patients, possibly via PRMTs due to increased proteolysis. There was no association of SDMA increase with stroke etiology in these two studies [85,86]. Controversially, in 363 ischemic stroke patients there was an increase in SDMA levels in patients with cardioembolic strokes but not in other stroke subtypes [25]. In concordance, in a cohort of 231 acute ischemic stroke patients SDMA levels were also elevated, when AF was detected in these patients [83]. Thus, it is noticeable that SDMA increase mostly occurs in the etiology of cardioembolic infarctions suggesting endothelial preconditions that provide levels in a certain dimension or even potentiate these levels in the event of stroke. However, cardioembolic strokes represent the most severe subtype, while the amount of destruction might further elevate SDMA levels. Interestingly, in a patient cohort of 88 ischemic stroke patients we found lower SDMA and higher ratios of Arg/ADMA, Arg/SDMA and ADMA/SDMA in patients with ESUS compared to patients with diagnosed AF as cardioembolic stroke etiology. SDMA and ADMA were determined at 7 days after stroke [46]. Most importantly, at follow-up of these patients at least one year after index stroke, SDMA values were almost stable over time (*p*  <  0.001; *r*  =  0.788) and still remained significantly higher in AF compared with ESUS-patients [116]. These results might indicate that indeed SDMA is rather related to the cardioembolic etiology than driven by the acute ischemic response (also see Section 5).

### Potential Mechanisms of Dimethylarginine Response after Ischemic Stroke

Obviously in ischemic stroke the acute and massive reduction of cerebral blood flow is to be made accountable for the immediate damage of neurons. It needs to be discussed that in the early stages of stroke pathology ADMA contributes to poor cerebral perfusion.

ADMA assumingly limits the cerebral perfusion in physiological conditions and in case of cerebrovascular injury by affecting the vascular tone and compliance. Nitric oxide synthase and NO represent the most important endogenous regulators of vasodilation in cerebral arterioles [117]. In an animal model in rats, administration of its endogenous inhibitor ADMA reduced the diameter of the basilar artery, while in rabbits the diameter of cerebral arterioles was reduced. Administration of Arg reversed this effect [118]. In mice administration of ADMA reduced response of cerebral arterioles to acetylcholine as vascular tone relaxation was restricted to 70%, whereas in transgenic mice with DDAH-1 overexpression relaxation of cerebral vessels as response to acetylcholine was not impaired [119]. In human middle cerebral arteries obtained from 26 autopsies administration of ADMA and L-NMMA impaired acetylcholine induced endothelial relaxation [120]. Additional administration of Arg revised this effect. Kielstein and colleagues administered ADMA in 20 healthy individuals over 40 minutes. In these individuals cerebral blood flow was detected to be significantly impaired, while systemic blood pressure was not altered [27]. These data suggest, that increased ADMA levels after the acute event of stroke might further reduce cerebral perfusion and thereby causes loss of penumbral tissue. One could even suggest that there is a link between increased ADMA levels and successful recanalization therapy, which had been investigated in acute stroke patient cohorts. Here, recanalization as reached by administration of recombinant tissue-type plasminogen activator (rtPA) and mechanical recanalization as performed by endovascular catheter therapy were investigated in separate clinical studies. In 41 acute ischemic stroke patients with large cerebral vessel occlusion undergoing mechanical recanalization, pretreatment levels of ADMA in patients with non-successful recanalization were higher than in successful therapy. However, in multivariate analysis there was no significant association between ADMA levels and the grade of recanalization [121].

Further evidence for a role of ADMA in acute medical recanalization treatment comes from 90 ischemic stroke patients of the German Multicenter EPO Stroke Trial. Patients had different treatment regimen, since they either received erythropoietin (EPO), placebo, rtPA + placebo or EPO + rtPA. Serum ADMA levels were observed from day 1 (pretreatment) until day 7 [122]. While ADMA levels increased in general, there was a significant reduction in the rtPA + placebo group compared with the placebo group. ADMA levels were correlated with outcome, although it remains unclear whether lower ADMA levels could contribute to favorable outcome. One might again suggest better reperfusion due to increasing NO levels after lack of endogenous inhibition by ADMA. Remarkably, rtPA is administered in a solution of rtPA and as carrier substance for stabilization of Arg. One could suggest, that early supplement of Arg in ischemic stroke might improve cerebral blood flow, while late supplement might lead to massive NO production ending in neurotoxic effects as caused by uncoupling and reactive oxygen species [123,124]. However, in 43 acute ischemic stroke patients with intravenous thrombolytic treatment using recombinant tissue-type plasminogen activator, pretreatment levels of ADMA were not associated with outcome [125]. In summary, so far, an independent influence of ADMA in recanalization treatment has been confirmed neither in mechanical nor in medical recanalization approaches.

During the acute stages of stroke, ADMA levels are increased by oxidative stress resulting in activation of PRMTs and inhibition of DDAH activity [126,127]. Here, ADMA might be a further generator of oxidative stress via uncoupling of eNOS and iNOS, which leads to synthesis of superoxides [126,128,129]. So far, studies are missing that show ADMA induced uncoupling of NOS in the cerebral circulation. After ischemic stroke, NO as synthesized by iNOS and nNOS is significantly increased. Extended NO levels possess neurotoxic properties since NO levels react with oxygen radicals to form the toxic compound peroxynitrite [130,131,132,133]. After ischemic stroke, expression of NOS isoforms differs in regard to temporal and spatial aspects. Mouse studies suggest that inhibition of iNOS and nNOS overexpression is neuroprotective, while inhibition of eNOS impairs cerebral blood flow and thereby could be detrimental for the penumbral tissue [134]. Herewith, effects of ADMA as assumingly non-selective NOS inhibitor might be protective and detrimental all at once. We suggest dose-dependent effects which might essentially be controlled by timing and the amount of NO produced. In macrophages NO-synthesis can be regulated by ADMA derived from endothelial cells [135]. However, it remains unclear if endothelial ADMA also regulates NO synthesis in neurons. 

Analysis of isoform specific inhibition of DDAH might clarify the diverse NO–DDAH–ADMA pathway after acute ischemic stroke, which would be urgently needed to identify any potential treatment targets. Recently, the importance of ADMA for NO-synthesis in human circulation has been challenged. While ADMA has been shown to effectively inhibit nNOS, the inhibitory potential of ADMA in regard to eNOS seems to be by far weaker [22,136]. This might at least indicate a lack of sufficient knowledge regarding the role of ADMA in patients. By influencing the inflammatory cascade after the acute event of stroke, ADMA might well enhance secondary brain injury, leading to worse patient outcome. A single therapeutic approach for lowering ADMA levels in an animal model was negative. In this transient middle cerebral artery occlusion (MCAO) model in mice infarction size did not differ in transgenic mice with DDAH-1 overexpression compared to wild-type animals [137]. However, neither ADMA levels nor DDAH-1 activity differed in these animals [137]. Recently, in a DDAH-1 knock-out rat model of MCAO, ADMA levels were increased while NO was reduced [138]. Administration of Arg in the knock-out group reduced neurological damage and increased levels of hypoxia inducible factor (HIF-1alpha) [138].

In different cell culture models, ADMA triggers production of pro-inflammatory cytokines [139,140]. In monocytes ADMA elevates the synthesis of TNF-alpha in a ROS/NF-kappaB dependent pathway [139]. In endothelial cells ADMA increases activation of NF-kappaB and phosphorylation of mitogen-activated kinases as well as levels of TNF-alpha and ICAM-1 [140]. ADMA enhances adhesion of polymorphonuclear neutrophils (PMN) on endothelial cells and triggers their degranulation [141].

In 58 ischemic stroke patients Chen and colleagues investigated the association of ADMA with mediators of inflammation [115]. The distinct temporal dynamics of ADMA levels after the event with peak values at time points when IL-6 levels already decreased may indicate that ADMA is unlikely to induce the inflammatory response. However, ADMA levels were associated with levels of IL-6 and CRP at several time points after stroke [115]. Molnar and colleagues detected an association of early ADMA and MCP-1 levels in 55 ischemic stroke patients [110]. Interestingly, in other conditions of acute inflammation, ADMA might rather be decreased in the acute setting, while levels elevate again when other inflammation markers decline [142].

Compared to ADMA data about the role of SDMA after acute stroke again are sparse. Experimental data point to a potential pathophysiological role of SDMA in acute stroke via modulation of NO levels, ROS and inflammatory processes. In experimental conditions concentrations of 1-10 mmol/L SDMA indirectly limits NO-synthesis by reduction of intracellular Arg uptake and renal tubular Arg absorption [143,144]. SDMA concentrations between 2-100µmol/L dose dependently enhanced ROS production in endothelial cells [82]. SDMA has been shown to be a weak nNOS inhibitor, while any data showing efficacy of SDMA for inhibition of other NOS isoforms are missing (for a review see Reference [136]). In monocytes SDMA increased ROS production by modulation of store-operated calcium channels [145]. Another study from the same group demonstrated that in monocytes SDMA increases activation of NF-kappaB and expression of TNF-alpha and IL-6 [146]. Certainly, these experimental conditions do not reflect those in stroke patients. 

In ischemic stroke patients, SDMA was significantly correlated with serum concentrations of the proinflammatory mediators MCP-1 and IL-6 [115]. Of note, SDMA and MCP-1 were correlated already at the early stage only hours after the acute event, while there were further correlations days after stroke between SDMA, MCP-1 and IL-6. Interestingly, Molnar and colleagues confirmed the early correlation of SDMA and MCP-1 [110]. SDMA and CRP were only correlated at later time points at 3 days after stroke [88]. In 43 ischemic stroke patients receiving IVT using rtPA, SDMA differed at each time point during the first week after stroke including pretreatment levels depending on clinical outcome [125]. However, multivariate analysis could not confirm an independent association between SDMA pretreatment levels and outcome in IVT. In summary, further studies are warranted to investigate if SDMA contributes in particular to secondary brain injury.

## 5. Arginine Derivatives as Markers of Stroke Etiology

### 5.1. Large Artery Atherosclerosis

The pathophysiologic link of Arg derivatives with atherosclerosis was discussed above. Cordts et al. [71] recently reported that ratios of hArg/ADMA and hArg/SDMA are associated with strokes due to AF or large artery atherosclerosis and are moreover predictive for prevalent stenosis of the internal carotid artery in three independent cohorts. Scherbakov and colleagues reported elevated ADMA levels in cardioembolic or large artery stroke [147]. In 262 ischemic stroke patients with intracranial atherosclerotic stroke, ADMA levels were increased compared to controls without stroke [148]. Interestingly, hArg/SDMA ratio showed a stronger relation with atherosclerotic burden compared with hArg/ADMA [71]. The degree of re-stenosis after carotid endarterectomy (CEA) has been found to be related to ADMA levels [149]. These results are clearly in line with the described relevance of Arg derivatives in the pathogenesis of atherosclerotic disease.

### 5.2. Small Vessel Disease

Tsuda et al. [150] reported an association of ADMA levels with occurrence of small vessel disease which was independent from classical cardiovascular risk factors. Moreover, ADMA correlated with the extent of cerebral leukoaraiosis. This finding was recently confirmed in a study investigating persons with asymptomatic white matter hyperintensities in whom inflammatory or coagulation disorders have been excluded [151]. Besides the association of ADMA to imaging measures of small vessel disease, a relation of ADMA levels with progression of cognitive impairment has been reported [152]. Furthermore, a role of ADMA has been implicated in rare microangiopathic diseases like Cerebral Autosomal Dominant Arteriopathy with Subcortical Infarcts and Leukoencephalopathy (CADASIL) [153]. Interestingly, dimethylarginines were shown to be positively correlated with Hachinsky Ischemic Score (HIS) and might therefore indicate vascular cognitive impairment [154]. In the study by Cordts et al. [71] hArg/ADMA and hArg/SDMA ratios enabled to discriminate lacunar from territorial strokes—an effect that was predominantly explained by higher hArg values in patients with small vessel disease compared with patients with other stroke etiologies. 

### 5.3. Cardioembolic Stroke and Embolic Stroke of Undetermined Source

The identification of the distinct etiology of ischemic stroke is crucial for its secondary prevention. Despite extensive diagnostic workup including cerebral, vascular und cardiac imaging as well as rhythm analyses a significant proportion of ischemic stroke remains unexplained. In 2015 Hart et al. [155] introduced the concept of the so called „embolic stroke of undetermined source“ (ESUS) which accounts for cryptogenic strokes with an embolic pattern in cranial imaging. ESUS accounts for approximately 16% of all ischemic strokes [156]. It is presumed that a significant part of ESUS will be due to subclinical paroxysmal atrial fibrillation (AF). As a result, two multicenter clinical trials (RE-SPECT ESUS [157] and NAVIGATE ESUS [158]) have been performed to test the hypothesis that ESUS patients might benefit from oral anticoagulation rather than platelet inhibition. However, both studies failed to prove this approach [157,158], indicating that proper workup for investigation of the individual stroke mechanism is still crucial for choosing the appropriate secondary preventive therapy. The same applies for concurrent stroke etiologies, like e.g. AF and high-grade carotid stenosis. In conclusion, novel biomarkers that support the diagnostics of stroke etiology would be advantageous for determining the appropriate individual therapy and consequently reduce the burden of recurrent strokes. 

Current data indicate that a diseased left atrium—recently referred as so called “atrial cardiopathy” [159]—might lead to enhanced thrombembolic risk independently of AF which is regarded as a mere symptom of the disease process [160,161]. Endothelial dysfunction mechanisms and subclinical atherosclerosis are closely related to the development of atrial cardiopathy in terms of a systemic disorder [162,163,164,165,166]. Hypothetically, endothelial dysfunction and arterial stiffness contribute to a higher cardiac afterload and consecutively to myocardial remodelling processes in the atrium [167,168]. Moreover, experimental and clinical studies implicate an important role of NOS signalling in AF [166]. As a result, Arg derivatives have been previously investigated as potential markers of AF. 

Schulze et al. [83] demonstrated that SDMA levels as well as the ratios between Arg/ADMA, Arg/SDMA and Arg/NMMA were significantly different in stroke patients with AF compared to those with sinus rhythm in a univariate analysis. Stamboul et al. [169] reported higher ADMA levels in patients who developed AF after myocardial infarction. A stepwise increase of ADMA concentrations has been shown in controls, in patients with lone AF and in patients with AF and other comorbidities [170]. This notion fostered further investigation relating ADMA and SDMA to permanent rather than paroxysmal AF in acute ischemic stroke [171]. Of note, AF patients exhibit a high proportion of cardiovascular risk factors. In an analysis of the Framingham cohort, Schnabel et al. [172] reported that Arg derivatives were not independently altered in AF patients after accounting for vascular risk factors. However, in another population-based study—the Gutenberg Health Study—we found a probable association of Arg derivatives with occurrence of AF, and moreover, independent correlations with electrocardiography and echocardiography-based measures of atrial disease [173]. Interestingly, SDMA, but not ADMA, was associated with left atrial dimension and P-wave duration in this analysis [173]. Cordts et al. [71] demonstrated a relation of hArg/ADMA and hArg/SDMA-ratios with cardioembolic stroke and stroke due to large artery disease.

In another study of patients with ischemic stroke SDMA and Arg/dimethylarginine ratios were significantly different between ESUS and AF in total as well as newly diagnosed AF [46], indicating a potential application in clinical practice. Furthermore, SDMA concentrations were shown to be associated with subsequently diagnosed AF after ESUS and correlate with left atrial volume index in ESUS patients and might therefore support identification of atrial cardiopathy in this stroke entity as recently reported by Ziegler et al. [116]. 

Interestingly, successful occlusion of the left atrial appendage led to a significant reduction of SDMA levels, while no changes concerning ADMA were observed [174]. Moreover, a polymorphism in the gene coding for AGXT2 which metabolizes ADMA and SDMA is associated with AF [175]. AGXT2 variants have been previously associated with circulating SDMA [176,177], heart rate variability [177] and stroke subtypes (lacunar vs. territorial infarctions) [176]. These data may indicate a pathogenic causal role of Arg derivatives and especially of SDMA in the development of AF. Of note, SDMA blood concentrations are largely dependent on renal function by which this marker is increasingly considered as potential measure for this purpose [178]. Conversely, renal insufficiency is a long-acknowledged factor in vascular diseases and AF [179]. Keeping in mind that AGXT2 is highly expressed in kidney cells [180], the SDMA metabolism is of high interest for future studies investigating the pathogenic link between this metabolite and renal and vascular disease. Of note, ADMA and SDMA might also be useful for risk prediction in anticoagulated AF-patients [79,181], as already discussed above.

Taken together, there are diverse and recent data underscoring the potential role of Arg derivatives as markers for identifying the underlying mechanism of ischemic stroke. Larger and prospective studies are needed to validate these findings.

## 6. Conclusions

Due to its physiological influence on vascular function, the Arg metabolism is of high interest in cerebrovascular diseases. ADMA is meanwhile confirmed as a biomarker of vascular risk, morbidity and mortality in a variety of large studies and meta-analyses. Although less studies are focussing on SDMA, this Arg derivative is currently emerging as another target and indeed might constitute characteristics divergent from ADMA, such as an association with cardioembolic stroke etiology. Besides the relation of Arg derivatives for risk and occurrence of ischemic stroke there is also a response of these metabolites after cerebral ischemia which further might contribute to secondary brain injury. In contrast, hArg is inversely associated with adverse events and mortality in cerebrovascular diseases and might constitute a modifiable protective risk factor. Taken together, Arg derivatives are promising diagnostic and therapeutic targets in diverse settings of cerebrovascular diseases. Future clinical studies are needed to validate the findings discussed in order to enable a translation into clinical practice.

## Figures and Tables

**Figure 1 ijms-21-01798-f001:**
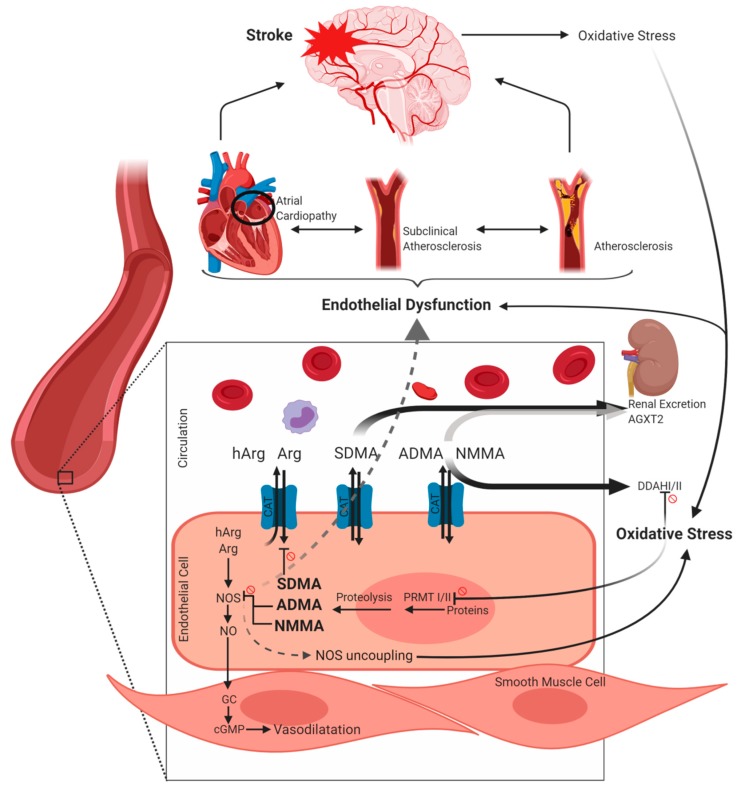
Overview on metabolism of arginine (Arg), homoarginine (hArg), asymmetric dimethylarginine (ADMA), symmetric dimethylarginine (SDMA) and monomethylarginine (NMMA) as well as putative links to cerebrovascular risk and disease. Prohibition signs besides lines refer to an inhibitory relation. For further explanation see Section 2. The figure was created using BioRender.

**Table 1 ijms-21-01798-t001:** Overview on selected clinical studies and meta-analyses evaluating ADMA in cerebrovascular diseases.

Reference	Sample Size (Population)	Biomarker Investigated	Comparison/Outcome	Adjusted HR/OR
Yoo et al. 2001 [24]	87 (IS patients and HC)	ADMA cutoff: 1.43 µmol/l	IS versus HC	OR: 6.05 (95% CI: 2.77–13.3)
Wanby et al. 2006 [25]	119 (IS patients and HC)	highest versus. lowest Arg/ADMA quartile	IS versus HC	OR: 0.28 (95% CI 0.11–0.72)
	89 (TIA patients and HC)	highest versus lowest ADMA quartile	TIA versus HC	OR: 13.1 (95% CI: 2.91–58.6)
Brouns et al. 2009 [85]	91 (IS patients and HC)	CSF ADMA	IS versus HC	NR
	45 (TIA patients and HC)	CSF ADMA	TIA versus HC	NR
Schulze et al. 2010 [83]	394 (IS patients)	ADMA	all-cause mortality	n.s.
Worthmann et al. 2011 [86]	67 (IS patients)	ADMA	clinical outcome	OR: 7.19 (95% CI: 1.73–29.82)
Rueda-Clausen et al. 2012 [87]	476 (IS patients and HC)	ADMA	IS versus HC	n.s.
Lüneburg et al. 2012 [84]	137 (IS patients)	ADMA	AE	n.s.
Molnar et al. 2014 [88]	55 (IS patients)	ADMA	all-cause mortality	n.s.
Willeit et al. 2015 [76]	8016 (IS patients and HC; meta-analysis)	highest vs. lower ADMA tertiles	IS versus HC	RR: 1.60 (95% CI: 1.33–1.91)
Emrich et al. 2018 [77]	528 (CKD patients)	ADMA	MACE	n.s.
Horowitz et al. 2018 [79]	4978 (anticoagulated AF patients)	ADMA	stroke/systemic embolism	HR: 1.19 (95% CI: 1.02–1.39)
			cardiovascular mortality	HR: 1.31 (95% CI: 1.18–1.46)
	4966 (anticoagulated AF patients)	ADMA	major bleeding	HR: 1.19 (95% CI: 1.07–1.34)

ADMA: asymmetric dimethylarginine; AE: adverse event; Arg: L-arginine; CES: cardioembolic stroke; CI: confidence interval; CKD: chronic kidney disease; CSF: cerebrospinal fluid; hArg: homoarginine; HC: healthy control; HR: hazard ratio; ICA: internal carotid artery; IS: ischemic stroke; LAA: large artery atherosclerosis; MACE: major adverse cardiovascular event; NR: not reported; n.s.: not significant; OR: odds ratio; RR: relative risk; SVD: small vessel disease; TIA: transient ischemic attack. HR and OR given derive from the appropriate multivariable analyses.

**Table 2 ijms-21-01798-t002:** Overview on selected clinical studies and meta-analyses evaluating SDMA in cerebrovascular diseases.

Reference.	Sample Size (Population)	Biomarker Investigated	Comparison/Outcome	Adjusted HR/OR
Brouns et al. 2009 [85]	91 (IS patients and HC)	CSF SDMA	IS versus HC	NR
	45 (TIA patients and HC)	CSF SDMA	TIA versus. HC	NR
Schulze et al. 2010 [83]	394 (IS patients)	highest versus lowest SDMA quartile	all-cause mortality	HR: 2.99 (95% CI: 1.64, 5.44)
Worthmann et al. 2011 [86]	67 (IS patients)	SDMA	clinical outcome	OR: 7.16 (95% CI: 1.67–30.69)
Lüneburg et al. 2012 [84]	137 (IS patients)	SDMA	AE	n.s.
Molnar et al. 2014 [88]	55 (IS patients)	SDMA	all-cause mortality	n.s.
Willeit et al. 2015 [76]	3132 (IS patients and HC; meta-analysis)	highest versus lower SDMA tertiles	IS versus HC	n.s.
Emrich et al. 2018 [77]	528 (CKD patients)	highest versus lowest SDMA tertile	MACE	HR: 2.678 (95% CI: 1.261–5.684)
Horowitz et al. 2018 [79]	4978 (anticoagulated AF patients)	SDMA	stroke/systemic embolism	n.s.
			cardiovascular death	HR: 1.40 (95% CI: 1.25–1.56)
	4966 (anticoagulated AF patients)	SDMA	major bleeding	HR: 1.41 (95% CI: 1.27–1.57)

ADMA: asymmetric dimethylarginine; AE: adverse event; Arg: L-arginine; CES: cardioembolic stroke; CI: confidence interval; CKD: chronic kidney disease; CSF: cerebrospinal fluid; hArg: homoarginine; HC: healthy control; HR: hazard ratio; ICA: internal carotid artery; IS: ischemic stroke; LAA: large artery atherosclerosis; MACE: major adverse cardiovascular event; NR: not reported; n.s.: not significant; OR: odds ratio; RR: relative risk; SVD: small vessel disease; TIA: transient ischemic attack. HR and OR given derive from the appropriate multivariable analyses.

**Table 3 ijms-21-01798-t003:** Overview on selected clinical studies evaluating hArg in cerebrovascular diseases.

Reference	Sample Size (Population)	Biomarker Investigated	Comparison/Outcome	Adjusted HR/OR
Choe et al. 2013 [91]	389 (IS patients)	1-SD increase in log hArg	all-cause mortality	HR: 0.79 (95% CI: 0.64–0.96)
	135 (IS patients)	1-SD increase in log hArg	AE	HR: 0.69 (95% CI: 0.50–0.94)
Pilz et al. 2014 [92]	606 (population based)	lowest versus higher hArg quartiles	cardiovascular mortality	HR: 4.20 (95% CI: 2·03–8·69)
Cordts et al. 2019 [71]	803 (IS patients)	hArg/ADMA hArg/SDMA	LAA/CES versus SVD/other	OR: 1.52 (95% CI: 1.12–2.06) OR: 2.01 (95% CI: 1.35–3.00
		hArg/ADMA hArg/SDMA	ICA stenosis versus no ICA stenosis	OR: 0.73 (95% CI: 0.55–0.97) OR: 0.69 (95% CI: 0.50–0.94)
Choe et al. 2020 [98]	394 (IS patients)	hArg/ADMA hArg/SDMA	all-cause mortality	HR: 0.75 (95% CI: 0.62–0.92) HR: 0.68 (95% CI: 0.54–0.85)
	135 (IS patients)	hArg/SDMA	AE	HR: 0.73 (95% CI 0.57–0.92)

ADMA: asymmetric dimethylarginine; AE: adverse event; Arg: L-arginine; CES: cardioembolic stroke; CI: confidence interval; CKD: chronic kidney disease; hArg: homoarginine; HC: healthy control; HR: hazard ratio; ICA: internal carotid artery; IS: ischemic stroke; LAA: large artery atherosclerosis; MACE: major adverse cardiovascular event; NR: not reported; n.s.: not significant; OR: odds ratio; RR: relative risk; SVD: small vessel disease; TIA: transient ischemic attack. HR and OR given derive from the appropriate multivariable analyses.

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
