# Peer review of "Arginine Derivatives in Cerebrovascular Diseases: Mechanisms and Clinical Implications"

_ijms, 2020, doi:10.3390/ijms21051798_

Round 1

Reviewer 1 Report

The review article presented by Grosse et al presents a comprehensive overview of the arginine derivatives (ADMA and SDMA) in CVD. I enjoyed reading this manuscript and only have some minor comments. 

1. The manuscript would benefit from a short introduction that includes background on stroke/CVD and the importance of understanding the role of arginine derivatives. This would help position the manuscript, but also help for readers not familiar with stroke

Author Response

We agree with the reviewer that an introduction from the clinical perspective would definitely be advantageous in order to underscore the potential impact of the biomarker findings described.

We accordingly added an introducing paragraph to the manuscript (p. 1, l. 28- p. 2, l. 48):

  1. Precision stroke medicine: on search for novel biomarkers

Stroke is globally the second leading cause of death and morbidity [1]. While stroke associated mortality decreased between 1990 and 2010, stroke prevalence, incidence as well as mortality rates again raised between 2010 and 2017 [2], despite optimized treatment options and intervention programs. Moreover, stroke burden is also increasing in young adults [3]. According to recent findings from the Global Burden of Disease study, the life-time risk to suffer stroke is about 25% starting at the age of 25 years [3]. Facing the challenges of this global cerebrovascular disease epidemic the need of biomarkers supporting individual stroke patient care in terms of precision medicine is becoming increasingly evident [4,5]. This holds true for estimating the individual risk for cerebrovascular diseases for primary preventive strategies but also for secondary prevention after the event. Stroke is a complex disease of diverse underlying risk factors and etiologies and current evidence underscores that a thorough individualized investigation of these conditions is needed for the purpose of an optimal treatment [4]. Thus, there are intense efforts in identifying appropriate imaging, genetic or blood biomarkers that are able to reflect the underlying pathophysiology and are useful for clinical decision making. L-arginine (Arg) derivatives may meet the conditions of such clinically interesting targets in cerebrovascular diseases. For this narrative review article, we conducted a comprehensive literature search through PubMed and selected original articles, reviews and meta-analyses on Arg and its derivatives in ischemic stroke, underlying risk factors and etiological diseases. We aim to provide an overview of the current evidence on how the Arg metabolism is involved in cerebrovascular pathophysiology and how Arg derivatives may constitute valuable biomarkers of risk, morbidity and mortality, as well as etiology of ischemic stroke.

Reviewer 2 Report

This is an extensive review discussing the role of dimethylarginines – ADMA and SDMA – in cerebrovascular disease.  The authors describe in great detail the role of dimethylarginines in vascular diseases such as atherosclerosis and provide several potential explanations for the role of ADMA in the initiation and evolution of vascular injury in cerebrovascular vasculature.  The review is comprehensive but a number of changes are required to improve reading.

A paragraph with clinical implications of the review would be extremely useful for the clinicians. How all these findings can be implemented in the clinical practice?

The text is too long and some parts – for example Homoarginine section – could be shortened.

A table summarizing the data of ADMA/SDMA in patients with stroke would add scientific value in the paper.

Research strategy is missing. The authors should describe the strategy through which they choose the articles included in the review (PMID:21800117).

Reference list should be enhanced by metaaanalysis exploring the role of dimethylarginines in RA (PMID:31376089) and publications underlying the interplay between inflammation and ADMA (PMID 25187642) as well as between ADMA and MTHFR polymorphisms (PMID:26599798) both of which represent risk factors for atherosclerosis. 

Author Response

We thank the reviewer for addressing this point which is similar to the suggestion by reviewer 1. We accordingly now added a paragraph (see point 1.1.) in order to improve the connection between the biomarker findings reported with the clinical perspective.

We agree with the reviewer that our article is extensive. Following the reviewer’s suggestion, we have shortened the Homoarginine section in order to improve readibility (p. 7, l. 238 – p. 9, l. 280):

“3.4. Homoarginine as marker and target in cerebrovascular diseases

A decade ago, hArg was studied in regard to cardiovascular and all-cause mortality [89]. In contrast to dimethylarginine derivatives, hArg levels were inversely associated with adverse events and mortality [90]. Most consistently, low hArg levels are associated with all-cause and cardiovascular mortality, which was shown in subjects referred for coronary angiography, in hemodialysis patients with diabetes mellitus, in stroke patients, but also population-based cohorts [89,91-94]. A recent meta-analysis confirmed the inverse association of hArg with all-cause mortality (HR 0.64 [0.57-0.73]) [95]. More specifically, low hArg levels were strongly associated with fatal strokes and revealed a trend to increase stroke risk [96,97]. In prospective studies of stroke patients, low hArg levels were independently associated with increased long-term all-cause mortality and short-term adverse events, respectively (Table 3) [91]. In addition to cerebrovascular patients, increased hArg levels (i.e. 1-SD log plasma hArg) were also associated with a risk reduction for major adverse cardiovascular events including stroke [93]. See table 3 for an overview on clinical studies evaluating hArg as biomarker in cerebrovascular diseases.

hArg has been implicated to play a role in vascular function [90,99-101]. Correspondingly, epidemiological studies revealed an inverse association of hArg with with aortic wall thickness and aortic plaque burden [93], an inverse correlation of hArg/ADMA ratio with aortic intima-media thickness [58] and a link between hArg/SDMA ratio with internal carotid artery stenosis and unfavorable outcome after stroke [71,98].

Most importantly, hArg supplementation confers indeed beneficial effects in vascular disease models. First, AGAT-deficient mice are devoid of hArg and revealed increased infarct sizes and an impaired cardiac contractibility, which were normalized upon hArg supplementation [91,102]. Furthermore, hArg supplementation attenuated detrimental effects of diabetic kidney damage, preserved systolic function in a model of coronary artery disease, reduced neointimal hyperplasia in balloon-injured rat carotids and attenuated post-myocardial infarction heart failure [91,103-106].

Although mouse studies revealed a causal link between hArg and vascular disease, the direct protective effect in humans remains to be established [107]. In a recent clinical trial, pharmacokinetic and -dynamic parameters were studied in healthy volunteers orally supplemented with 125mg hArg or placebo daily for 4 weeks using a cross-over design [108]. Supplementation was well tolerated and increased hArg levels by 7-fold without any alteration of vascular or neurological parameters [108,109]. Currently, a randomized placebo-controlled trial studies the administration of oral hArg in acute stroke patients with low hArg levels (https://www.clinicaltrials.gov. Unique identifier: NCT03692234). Future studies will answer the question, if hArg deficiency is a modifiable protective cerebrovascular risk factor.”

Indeed, tables summarizing relevant data would be a substantial improvement of our article. According to the reviewer’s suggestion, we added three tables for each ADMA (p. 6, ll. 212-219), SDMA (p. 7, ll. 229-236) and hArg (p. 8, ll. 261-267) presenting the studies discussed in order to provide an overview on the current evidence of these markers in cerebrovascular diseases.

We thank the reviewer for raising this question. We added this information in the introducing paragraph as follows (p. 1, l. 43 – p. 2, l. 48):

“For this narrative review article, we conducted a comprehensive literature search through PubMed and selected original articles, reviews and meta-analyses on Arg and its derivatives in ischemic stroke, underlying risk factors and etiological diseases. We aim to provide an overview of the current evidence on how the Arg metabolism is involved in cerebrovascular pathophysiology and how Arg derivatives may constitute valuable biomarkers of risk, morbidity and mortality, as well as etiology of ischemic stroke.”

We agree that the link between ADMA, inflammation and atherosclerosis was missing in our article. According to the reviewer’s suggestion we have included the mentioned publications in a new paragraph addressing the link between ADMA and inflammation (p. 4, ll. 124-131):

“Progression and vulnerability of atherosclerotic lesions is driven by inflammatory mechanisms [33]. ADMA may be increased by systemic inflammation and subsequently lead to endothelial dysfunction in patients with coronary heart disease or rheumatoid arthritis (RA) [34]. The link of ADMA and inflammation in RA patients is meanwhile confirmed by different studies and meta-analyses [35-37] suggesting that ADMA may constitute as vascular risk marker in this vulnerable patient collective. This is further supported by a study in RA patients demonstrating an association of ADMA with homocysteine and with a methylenetetrahydrofolate reductase (MTHFR) polymorphism, which are implicated in atherosclerosis [38].”

Round 2

Reviewer 2 Report

The authors have adequately addressed the comments and the paper is eligible for publication.